Manuscript prepared for Atmos. Meas. Tech.
with version 2014/09/16 7.15 Copernicus papers of the LATEX class copernicus.cls.
Date: 11 December 2018

# A new MesosphEO dataset of temperature profiles from 35 to 85 km using Rayleigh scattering at limb from GOMOS/ENVISAT daytime observations

Alain Hauchecorne[1], Laurent Blanot[2], Robin Wing[1], Philippe Keckhut[1],
Sergey Khaykin[1], Jean-Loup Bertaux[1], Mustapha Meftah[1], Chantal Claud[3], and
Viktoria Sofieva[4]

[1]LATMOS/IPSL, UVSQ Université Paris-Saclay, Sorbonne Université, CNRS, Guyancourt, France
[2]ACRI-ST, Sophia Antipolis, France
[3]LMD, Ecole Polytechnique, CNRS/INSU, Palaiseau, France
[4]FMI, Helsinki, Finland

*Correspondence to:* Alain Hauchecorne (alain.hauchecorne@latmos.ipsl.fr)

**Abstract.** Given that the scattering of sunlight by the Earth's atmosphere above 30-35 km is primarily due to molecular Rayleigh scattering, the intensity of scattered photons can be assumed to be directly proportional to the atmospheric density. From the measured relative density profile it is possible to retrieve an absolute temperature profile by assuming local hydrostatic equilibrium, the perfect gas law, and an a priori temperature from a climatological model at the top of the atmosphere. This technique has been applied to Rayleigh lidar observations for over 35 years. The GOMOS star occultation spectrometer includes spectral channels to observe daytime limb scattered sunlight along the line of sight to a reference star. GOMOS Rayleigh scattering profiles in the spectral range of 420-480 nm have been used to retrieve temperature profiles between 35-85 km with a 2-km vertical resolution. Using this technique, a database of more than 309,000 temperature profiles has been created from GOMOS measurements.

A global climatology was constructed using the new GOMOS database and is compared to an external model. In the upper stratosphere the external model is based on the ECMWF re-analysis and the agreement with GOMOS is better than 2 K. In the mesosphere the external model follows the MSIS climatology and 5 to 10 K differences are observed with respect to the GOMOS temperature profiles. Comparison to nighttime collocated Rayleigh lidar profiles above the south of France show some vertical structured temperature differences which may be partially explained by the contributions of the thermal diurnal tide.

The equatorial temperature series shows clear examples of mesospheric inversion layers in the temperature profiles. The inversion layers have global longitudinal extension and temporal evolution, descending from 80 to 70 km over the course of a month. The climatology shows a semi-annual

temperature variation in the upper stratosphere, a stratopause altitude varying between 47 and 54 km, and an annual variation in the temperatures of the mesosphere. The technique to derive temperature profiles from Rayleigh limb scattering can be applied to any other limb-scatter sounder, providing

that the observations are in the spectral range 350-500 nm. Due to the simplicity of the principles involved, this technique is also a good candidate for a future missions where constellations of small satellites are deployed.

## 1 Introduction

The middle atmosphere (MA: stratosphere and mesosphere, 12 to 90 km altitude) is a transition

region between the troposphere, which is heavily influenced by anthropogenic activity, and the upper atmosphere (thermosphere and ionosphere) at the edge of the space and strongly impacted by solar activity. The MA is a unique environment for fundamental research as it is subject to the conjugated influence of climate change, due to anthropogenic activities, and natural solar driven variability.

The increase of GHGs (Green House Gases) induces a global warming at the surface and of the

troposphere at large but, also affects a global cooling in the MA (e.g. (Keckhut et al., 2011)). The cooling occurs as thermal infrared radiation emitted by GHGs escapes directly into space due to the low optical thickness of the atmosphere above.

The mesosphere is a region where temperature and wind observations are sparse or not well resolved. Recent studies have demonstrated the role of MA dynamics in both tropospheric weather

and climate ((Baldwin and Dunkerton, 2001); (Shaw et al., 2014); (Charlton-Perez et al., 2018)) as well, weather and climate-chemistry models are currently moving towards a more comprehensive representation of the MA ( (Beagley et al., 2000); (Baldwin et al., 2003); (Hardiman et al., 2010)). Atmospheric observations in this region can also be used as a benchmark for climate change studies due to the MA has a high sensitivity to the increase of GHGs and to the external solar forcing.

Furthermore, technical and scientific questions relating to applications such as the re-entry of space and sub-orbital vehicles, the impact of meteors on the atmosphere, and infrasound propagation modelling in the atmosphere, require an accurate understanding of the mesospheric mean state and its variability at different scales.

There are insufficient observations of the temperature in the upper portions of the given that the

upper limit of radiosondes is about 30 km. GNSS (Global Navigation Satellite System) Radio-Occultation technique provides accurate measurements of temperature up to about 35 km with high vertical resolution. Nadir viewing satellite sensors making observations in the thermal infrared (e.g. SSU: Stratospheric Sounder Unit) and at microwave wavelengths (e.g. AMSU: Advanced Microwave Sounding Unit) extend measurements of brightness temperature into the upper stratosphere

(around 45 km) but often have very broad vertical weighting functions ($\approx$ 10 km). These coarsely resolved operational satellites provide the only temperature observations assimilated into NWP (Nu-

merical Weather Prediction) models. Limb viewing satellite sounders such as MLS (Microwave Limb Sounder on the Aura satellite) and SABER (Sounding of the Atmosphere using Broadband Emission Radiometry on the TIMED mission), provide temperature profiles up to the upper meso-

sphere with a good vertical resolution. However these datasets are not assimilated into the NWP models because MLS and SABER are not operational meteorological satellites.

The scattering of sunlight (near the UV and visible wavelengths) by the Earth atmosphere above the top of the stratospheric layer (30-35 km) is solely due to Rayleigh scattering by atmospheric molecules. The elastic scattering intensity is directly proportional to the atmospheric density. It is

thus possible to retrieve an absolute temperature profile using the hydrostatic equation and the perfect gas law. The temperature is initialised at the top of the measurement profile from a climatological model. This inversion technique has been applied to Rayleigh lidar observations for more than 40 years (Hauchecorne and Chanin, 1980). Approximately 10 Rayleigh lidars are operated routinely in the NDACC (Network for the Detection of Atmospheric Composition Changes). These ground

stations are limited in number (approximately 10 distributed globally) but routinely produce local observations of the atmospheric temperature profile between 30 and 80-90 km with a good accuracy and vertical resolution (Keckhut et al., 2011). They have been used for trend analysis ((Hauchecorne et al., 1991); (Keckhut et al., 1995); (Li et al., 2011)) and or validation of satellite data and identification of possible biases and trends due to orbital changes and instruments ageing ((Funatsu et al.,

2008); (Keckhut et al., 2015); (Funatsu et al., 2016)).

The observation from space of Rayleigh scattering at the atmospheric limb during daytime may be also used to derive density and temperature profiles in the upper stratosphere and mesosphere (US-M). This technique has been applied by (Clancy et al., 1994) who derived temperature profiles from 40 to 92 km for the period 1982-1986 using Solar Mesosphere Explorer bright limb observa-

tions at 304, 313 and 442 nm. (Shepherd et al., 2001) determined temperature profiles from 65 to 90 km during the period of March 1992 - January 1994 by analysing WINDII/UARS data at 553nm. More recently (Sheese et al., 2012) retrieved temperature profiles using OSIRIS/Odin bright limb observations at 318.5 and 347.5 nm in the altitude range 45-85 km. In the frame of the ESA funded MesosphEO project, a new dataset of temperature profiles in the altitude range 35-85 km was cre-

ated from the analysis of GOMOS/ENVISAT bright limb observations in the spectral band 420-480 nm. A dataset of more than 309,000 profiles from June 2002 to April 2012 is now available for climatological and dynamical studies.

The paper is organised as follows: In Sect. 2, the principle of the method is explained and the data processing is described. Section 3 is dedicated to the validation of the GOMOS temperature

profiles using Rayleigh lidar observations from OHP. Section 4 presents the first scientific results with a focus on the evolution of equatorial temperature profiles. Finally, a summary is given in Sect. 5.

## 2 Principle and data processing

### 2.1 Method

GOMOS (Global Ozone Monitoring by Occultation of Stars), on board the European Space Agency ENVISAT (ENVIronmental SATellite) platform, was the first operational space instrument dedicated to the study of the middle atmosphere using stellar occultation technique. A description of the instrument as well as an overview of the main scientific results is given in (Bertaux et al., 2010). GOMOS observes the spectrum of a star at various angles during its occultation by Earth's atmosphere.

The atmospheric transmission spectrum is equal to the ratio between the star spectrum absorbed by the atmosphere and the reference star spectrum which is measured outside the atmosphere. Given a particular atmospheric transmission spectrum, atmospheric constituents can be identified by their unique absorption features. Given that the stellar reference spectrum is measured at the beginning of each occultation cycle and that GOMOS is independent of any radiometric calibration, we can

consider GOMOS to be a self-calibrated instrument. Furthermore the stellar occultation technique allows a perfect knowledge of the tangent altitude, depending only on the geometry of the light path between the star and the satellite. The 250-680 nm spectral domain is used for the determination of $O_3$, $NO_2$, $NO_3$ relative density profiles as well as for profiles of aerosols from the upper troposphere to the mesosphere (Kyrölä et al., 2010). In addition, two high spectral resolution channels

centred at 760 nm and 940 nm allow for the measurement of $O_2$ and $H_2O$, respectfully. In order to remove the background signal due to the sunlight scattered by the atmosphere, two background spectra are taken just above and below the location of the star. We will refer to these two reference spectra as as upper and lower spectra. In this study we only use background spectra during daytime (bright limb occultations). Bright limb spectra have been used to derive vertical profiles of ozone

during daytime (Tukiainen et al., 2011). (Bertaux et al., 2010) identified seven possible methods to determine temperature profiles from GOMOS data. Among the various methods, the two most promising are the vertical inversion of the Rayleigh scattering profile at limb and the time delay between blue and red scintillations due to chromatic refraction. The two methods are complimentary and this article presents details and results from the first method while an improved algorithm for the

second method is presented in (Sofieva et al., 2018). The Rayleigh scattering method covers the altitude range 35-85 km during daytime and the chromatic refraction method covers the altitude range 15-32 km during nighttime. For each daytime occultation a vertical profile of bright limb light is calculated by averaging over three 20-nm spectral bands, 420-440 nm, 440-460 nm and 460-480 nm in the upper and lower background spectra. Above 35 km the scattering of sunlight by stratospheric

aerosols is negligible and the measured signal at 420-480 nm is only due to the Rayleigh scattering by atmospheric molecules. Given that, at these wavelengths absorption due to ozone and other trace gases is negligible, the number of scattered photons is assumed to be directly proportional to the atmospheric density. Figure 1 shows an example of a limb scattering profile in three spectral bands.

The decrease of the Rayleigh scattering signal due to the exponential decrease of the atmospheric
density is seen up to about 70 km. Above this altitude the contribution of the measurement noise becomes more important but the Rayleigh signal can be exploited up to at least 90 km after removing this noise.

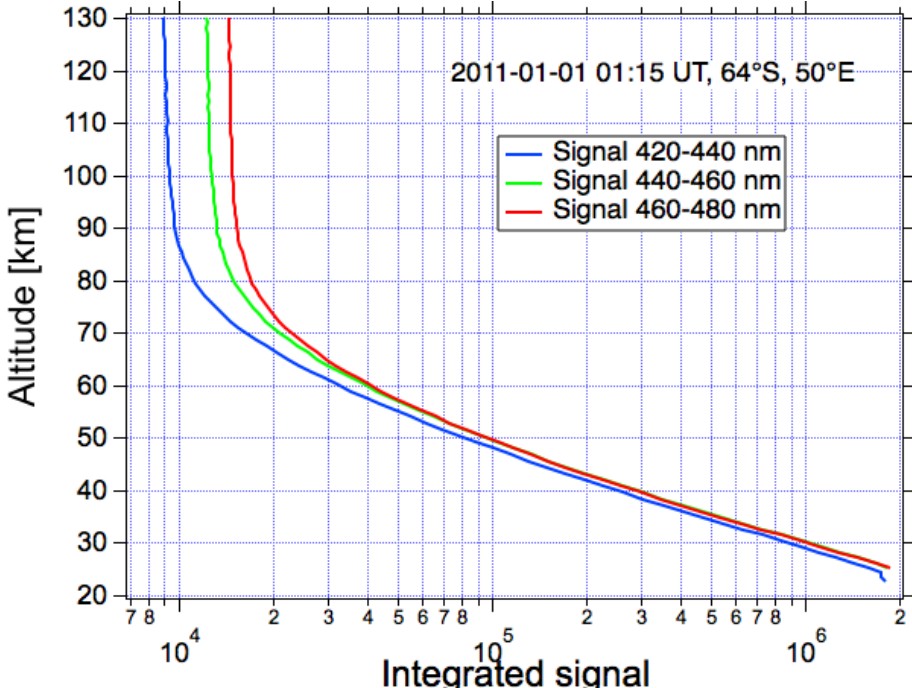

Figure 1: Spectrometer signal integrated in the three 20-nm spectral ranges for one occultation on 1st January 2011 (star ID =3, orbit number = 46209), lower background spectrum.

### 2.1.1   Data processing

For this study we used the full Level 1 GOMOS database between June 2002 and April 2012 which
comprises more than 418,000 bright limb occultations. A screening is conducted to select the measurements used for the generation of the temperature profiles:

- Occultations with a solar zenith angle larger than 84° are eliminated to avoid spectra with too much absorption between the sun and the limb.

- Profiles, which do not cover the altitude range between 35 and 125 km, are not considered.
The lower limit is set to retrieve a temperature profile covering the full altitude range 35-85 km and the upper limit is set to have enough data at the top of the profile to correctly estimate the measurement noise.

– Occultations with the presence of Polar Mesospheric Clouds (PMCs) are also removed. PMCs detection is based on the algorithm described by (Pérot et al., 2010). After this screening 309,341 occultations are selected.

### 2.1.2 Processing one occultation

For each spectrum in the upper and lower background bands of the GOMOS A2 spectrometer (400-680 nm), the signal is integrated in 3 spectral ranges, 420-440 nm, 440-460 nm and 460-480 nm to obtain 6 spectral profiles as a function of tangent altitude. After removing contributions from stray light and detector noise, which are estimated at altitudes above 110 km and extrapolated down to lower altitudes, a vertical inversion is performed using an onion peeling method. The resulting 6 profiles of Rayleigh scattering versus altitude are assumed to be proportional to the atmospheric density. The algorithm to retrieve temperature profiles is very similar to the Rayleigh lidar algorithm described in detail in (Wing et al., 2018a). The temperature is computed by downward integration of the hydrostatic equation 1 assuming the perfect gas law 2:

$$dP(z) = -\rho(z)g(z)dz \tag{1}$$

$$P(z) = \frac{R\rho(z)T(z)}{M} \tag{2}$$

where z is the altitude, P is the pressure, T the temperature, $\rho$ the atmospheric density, g the gravity, R the perfect gas constant (R=287.06 $\frac{J}{K \cdot kg}$), and M the air molar mass (M=28.96). The initialisation of the pressure at the top of the profile is made near 95 km assuming that the mean temperature in the layer 85-95 km is equal to the temperature of the NRLMSISE-00 climatological model (Picone et al., 2002). For each occultation, 6 individual temperature profiles are retrieved corresponding to the three selected wavelength intervals. Examples of these profiles can be seen in the upper and lower panels of Fig. 2. For the following analysis, we use only the median of these six temperature profiles as the temperature profile corresponding to the occultation, and the dispersion (1 standard deviation) interval of the 6 individual profiles) as an estimation of the uncertainty.

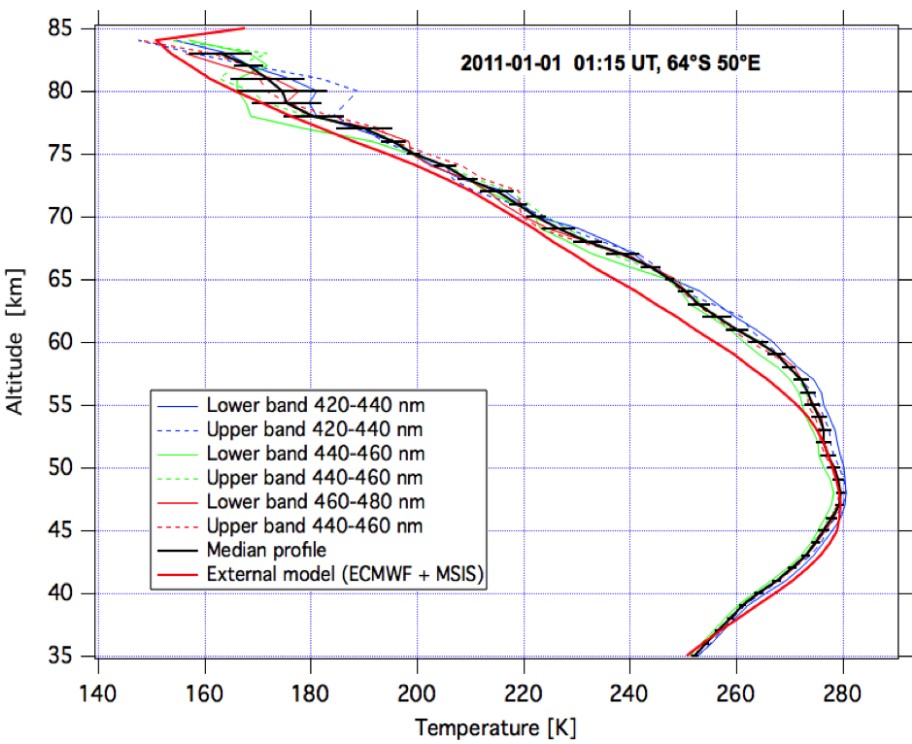

Figure 2: Temperature profiles processed for the same occultation as in Fig. 1. The horizontal bar indicates the dispersion (1 standard deviation) between the 6 individual profiles.

## 3  Validation using Rayleigh lidar observations

A validation exercise has been made using the Rayleigh lidar located at Observatoire de Haute Provence (OHP; 43.9° N, 5.7° E). This lidar has been part of the Network for Detection of Atmospheric Composition Change (NDACC; http://www.ndsc.ncep.noaa.gov/) since its creation in 1991 and has participated in several satellite validation experiments for instruments on board UARS satellite ( (Fishbein et al., 1996); (Gille et al., 1996); (Hervig et al., 1996); (Singh et al., 1996); (Wu et al., 2003); (Keckhut et al., 2004)), and more recently for MLS-Aura and SABER-TIMED (Wing et al., 2018b). For the present study 554 collocated GOMOS profiles were selected in a geographic region around OHP (40° N, 9° E); (48° N, 21° E). The nightly mean lidar profiles are smoothed down to a 3-km vertical resolution for comparison with the GOMOS profiles. A maximum of 12h difference between GOMOS and lidar measurements was accepted for the time coincidence. When several GOMOS profiles reached the coincidence criteria, as shown in Fig. 3, the median profile was used for the statistical comparison.

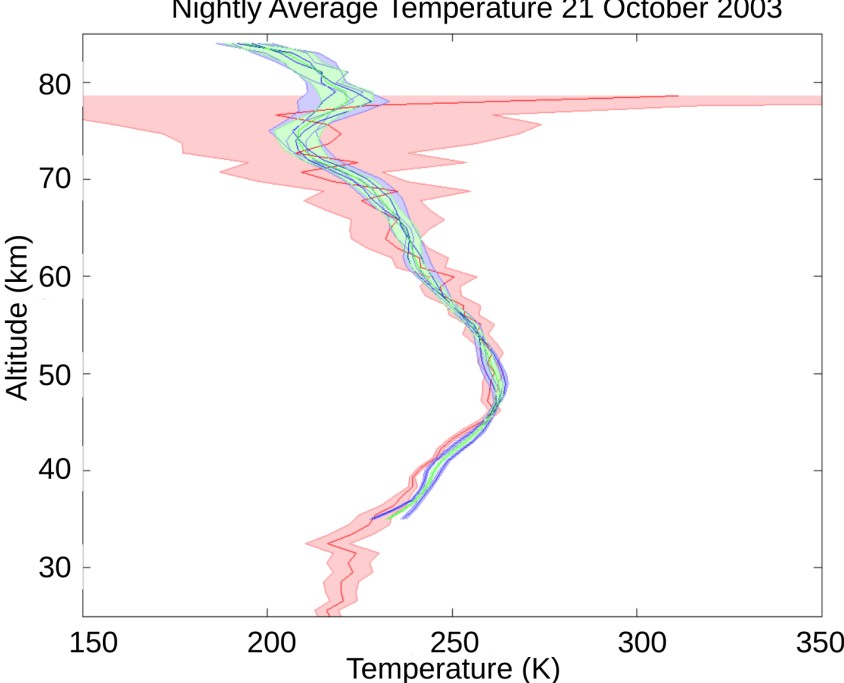

Figure 3: An example of a comparison between a Rayleigh lidar profile at OHP on 21st October 2003 (in red) and two collocated GOMOS profiles selected using the co-location criteria (in blue and green). When two or more GOMOS profiles are selected, the median profile is used for the statistical comparison. For the lidar profile (red), each individual GOMOS profile (blue) and the median GOMOS profile (green), the standard deviation of its uncertainty is represented by the shaded area.

The statistical median difference between the OHP lidar and GOMOS temperatures (Fig. 4) is close to zero below 46 km. There is a negative relative bias between 46 and 73 km with a maximum difference of -5 K between 55 and 60 km and positive relative bias above 73 km with a maximum difference of +7 K at 85 km. The dispersion of the differences remains relatively constant with altitude and has an approximate value of $\pm$ 5 K over the entire altitude range. The positive relative

bias in the upper part of the profile may be at least partially due to a warm bias in OHP temperature above 75 km as reported by (Wing et al., 2018b) using a comparison with SABER-TIMED. Below 75 km, the alternation between positive and negative relative biases with altitude may indicate a contribution of the atmospheric thermal tides, as the temperature measurements are not obtained simultaneously. The tidal effect has been previously observed in comparisons between measurements

obtained at different solar times (Wild et al., 1995); (Keckhut et al., 1996); (Keckhut et al., 2015)). GOMOS measurements above OHP are performed during daytime at around 11:00 am solar time while lidar operations are conducted during the first part of the night for several hours, with an estimated average mid-sequence time around 21:00 solar time.

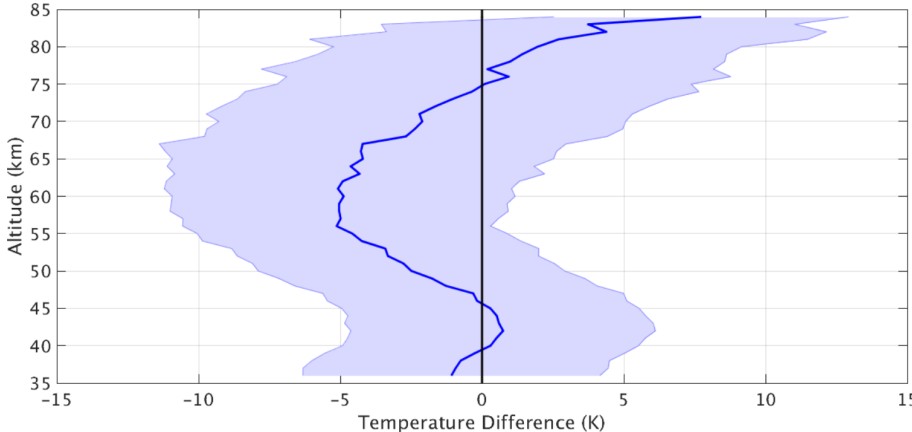

Figure 4: Statistical median temperature difference between OHP lidar and GOMOS temperature profiles (lidar minus GOMOS temperature). The shaded area represents the dispersion of the differences (one standard deviation).

To evaluate the potential effect of the thermal tides, tidal characteristics above the lidar site have been extracted from the Global Scale Waves Model (GSWM; (Hagan et al., 1999)) and used to provide an estimate of the tidal contribution to the observed temperatures differences. The model has been optimised to provide the migrating thermally forced tides on a global scale throughout the atmosphere on a monthly mean basis. The amplitude and phase of the diurnal and the semi-diurnal components can be calculated from the outputs of the GSWM-00 tidal model (http://www.hao.ucar.edu/modeling/gswm/gswm.html), which is an extension of the GSWM-98 (Hagan et al., 1999). This model has previously been used in comparisons with observations. While the vertical shape of the observed lidar-GOMOS relative temperature bias is well reproduced using this model, the amplitude is often smaller than those reported by (Raju et al., 2010). In this study, the amplitude (Fig. 5, left panel) and the phase (Fig. 5, middle panel) of the diurnal component of the tides have been extracted from the GSWM for the 45°N latitude during the month of August). In the summer, the middle atmospheric component of the diurnal tide is dominant and the expected difference between the lidar and GOMOS temperatures is represented in Fig. 5, right panel. In the middle mesosphere we observe a +3 K difference while in the vicinity of the mesopause, we note a reverse effect of -3 K. The expected tidal contribution does not fully reproduce the observed temperature difference between the OHP lidar and GOMOS but, considering uncertainties associated with the amplitude and phase of the tidal effect, and the fact that non-migrating tides were not taken into account, it appears that at least some part of the observed differences may be explained by local time differences. Further work would be needed to confirm this hypothesis. The comparison of OHP lidar temperature profiles with MLS-Aura and SABER-TIMED indicated systematic differences and suggested non-linear distortions in the satellite altitude retrievals (Wing et al., 2018b). In order to better understand

these differences, we plan to compare in a future work our new GOMOS temperature dataset with MLS and SABER.

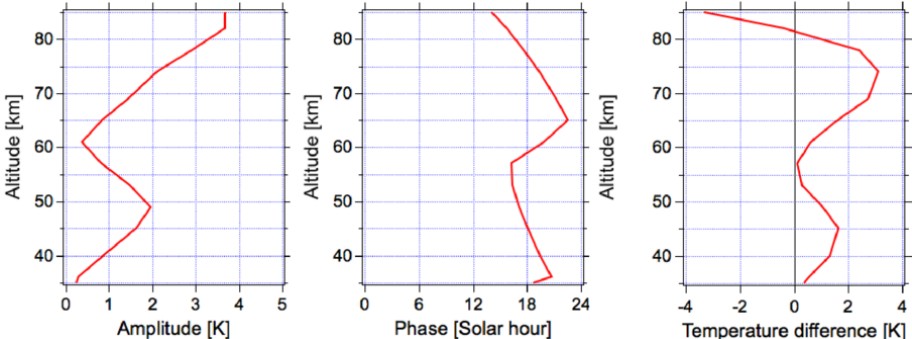

Figure 5: Amplitude (left panel) and phase (time of the maximum temperature; right panel) of the diurnal tides extracted from the GSWM above 45°N for August. Temperature differences (lidar minus GOMOS; right panel) expected from diurnal atmospheric tides as simulated by the GSWM-00 model.

## 4 First scientific results

The monthly climatology of GOMOS temperature has been built by averaging the data into 10° bins
from 80°S to 80°N. For each monthly latitude bin, the average value is only considered if at least 15 valid profiles are kept. The results are presented in fig. 6. At the stratopause the warmest temperatures are observed at the North Pole from April to September and at the South Pole from November to January, the equatorial stratopause temperature show a relative maximum throughout the year. As expected, the coldest temperatures are observed in the upper mesosphere at high latitudes during the
summer months, from May to August in the northern hemisphere and from November to February in the southern hemisphere. The deep temperature minimum in the summer mesopause is due to adiabatic cooling of ascending air.

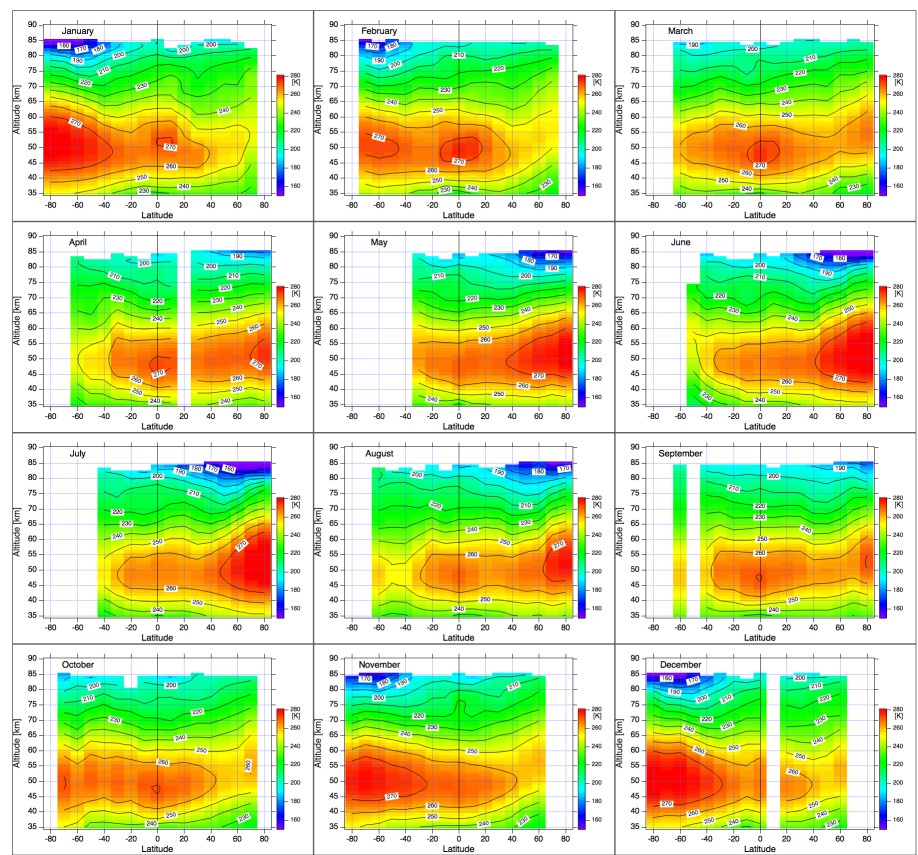

Figure 6: Monthly climatology of GOMOS Rayleigh temperature. The data are averaged over 10°
latitude bins.

In order to better visualise the main features of the GOMOS climatology, the temperature difference between the climatology and an external model are represented in Fig. 7. The external model used for processing the GOMOS data relies on the retrieval of different atmospheric species as described by (Kyrölä et al., 2010). For each occultation the external atmospheric profile is built by using ECMWF analysis up to 1 hPa (about 48 km) with a smooth transition to NRLMSISE-00 climatological model above 1 hPa, preserving the hydrostatic equilibrium at all altitudes.

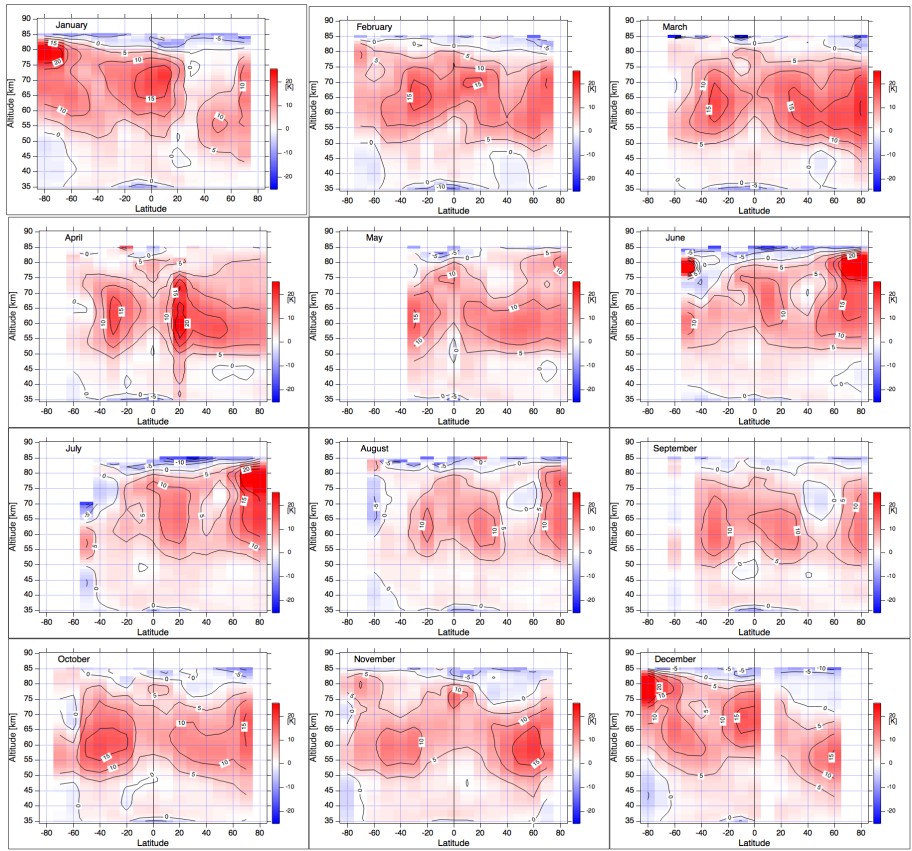

Figure 7: Monthly climatology of the temperature difference between GOMOS and the external (ECMWF+MSIS). Data are averaged over 10° latitude bins.

Figure 8 shows the average temperature difference between GOMOS and the external model,
averaged over all latitudes and months. Below 48 km, where the external model is based on ECMWF analysis, the agreement is very good and is almost always better than 5 K and on average better than 2 K. The only exception is at 35 km in the equatorial region where GOMOS presents a cold bias compared to the model, in particular from January to May a cold bias of about -10 K is seen. We attribute this cold bias to a contamination of the Rayleigh scattering profile by Mie scattering due
to the presence of aerosols in the lower stratosphere, these aerosols may reach altitudes of 35 km at the equator (Vernier et al., 2009). Above 48 km the external model is driven by NRLMSISE-00 and between 48 and 80 km the GOMOS temperature is warmer than the external model. Near 60 km the temperature difference is on average +10 K. Above 80 km GOMOS is colder than the external model.

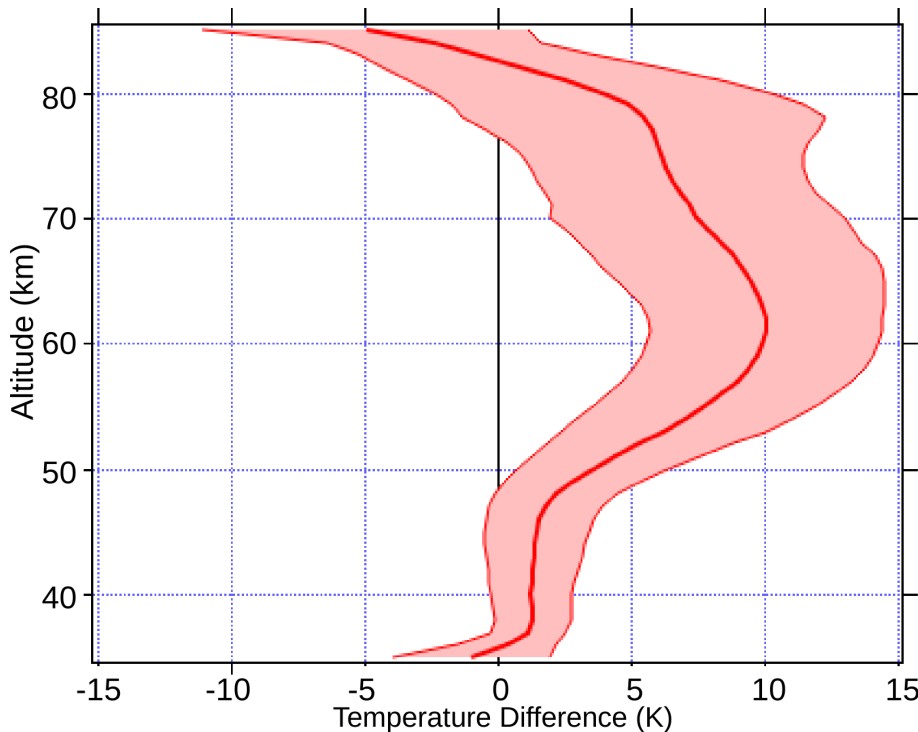

Figure 8: The mean difference between GOMOS Rayleigh temperatures and the external model temperatures as a function of altitude. The standard deviation of the difference is shaded.

An interesting characteristic of the measurement which arises from the geometry of observation is that for a given line of sight, parallel to the Earth's polar axis, the tangent point in the atmosphere is exactly at the Equator. The occultation of the Polar Star, at approximately 89.5° declination, provides a year round tangent reference point between 0.8°S and 0.8°N in bright limb conditions. More than 22,000 occultations of the Polar Star have been performed during the 10 years of the ENVISAT

record, providing a quasi-continuous survey of the temperature evolution at the Equator (Fig. 9, left panel). The temperature at the stratopause exhibits a semi-annual variation while in the mesosphere we observe the descent of cold layers from 80 to 70 km over the course of one month. Several intense cold layers occurred in April-May 2007 and the vertical profile for the first week of May ((Fig. 9, right panel) shows that this cold layer corresponds to a so-called mesospheric inversion layer (MIL)

in the vertical temperature profiles.

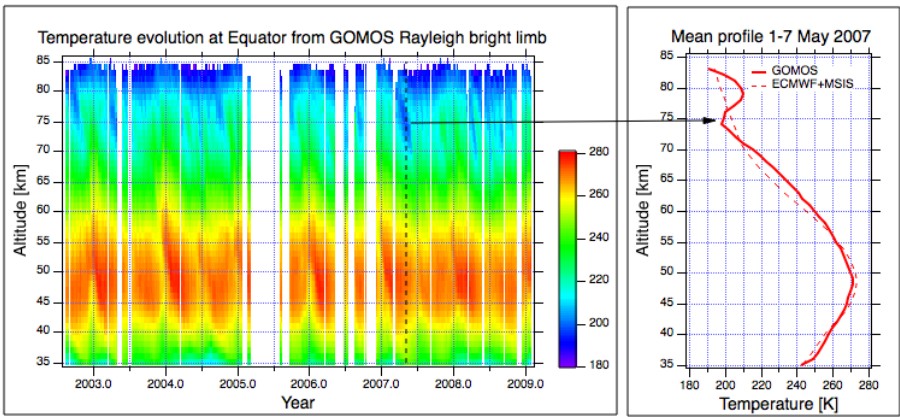

Figure 9: Left) Evolution of the weekly averaged temperature profile at Equator obtained using all occultations of the polar star with a tangent point latitude always situated in the interval 0.8°S-0.8°N. Right) Vertical weekly mean profile beginning of May compared to the GOMOS external model.

MILs have been observed by rocketsondes (Schmidlin, 1976) and Rayleigh lidars at middle latitudes ((Hauchecorne et al., 1987); (Duck et al., 2001)), high latitudes (Cutler et al., 2001) and low latitudes (Ratnam et al., 2003). Satellite observations showed the global extend of MILs ((Leblanc and Hauchecorne, 1997); (Fechine et al., 2008); (Gan et al., 2012)). Several explanations have been

proposed to explain the formation of MILs including gravity wave breaking ((Hauchecorne and Maillard, 1990)), planetary wave structure (Salby et al., 2002) and thermal tides (Meriwether et al., 1998). Explanations of the long duration and the global longitudinal extend of the observed equatorial MILs are beyond the scope of this paper and will be the topic of a future publication.

Polar Star profiles have been used to build a seasonal climatology of equatorial temperatures Fig.

10. In the upper stratosphere, the dominant feature is the semi-annual temperature oscillation which has maxima during the equinoxes (February to April and September-October) and minima during the solstices (June-July and December). The altitude of the stratopause, taken at the altitude with warmest temperature, varies between 47 and 54 km during the year with a primary maximum in December-January and a secondary maximum in July. In the mesosphere the temperature evolution

is dominated by the annual oscillation which has a maximum in December-January, corresponding to the period with an elevated stratopause, and a long minimum from April to October.

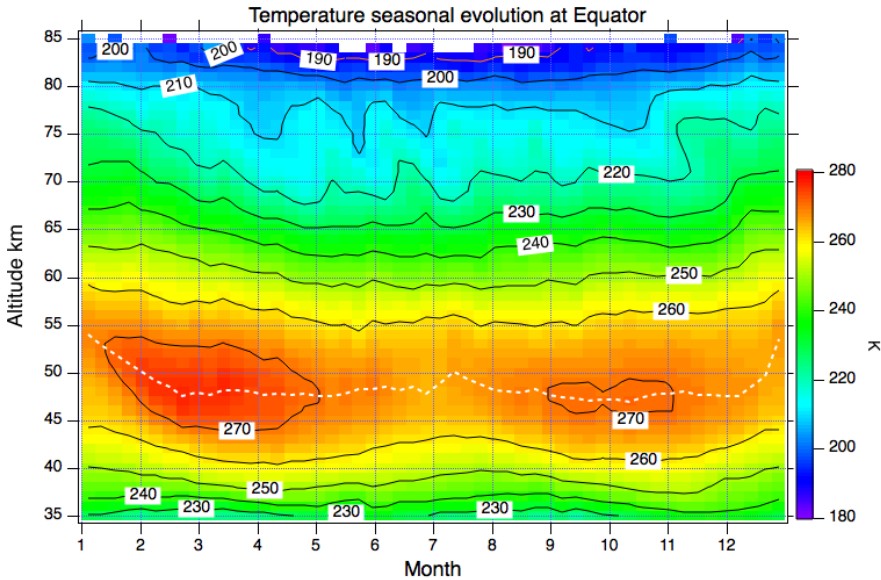

Figure 10: Seasonal evolution of the equatorial temperature derived from temperature data presented in Fig. 2 The altitude of the stratopause is indicated by the white dotted line.

## 5 Conclusions

A database of more than 309,000 temperature profiles from 35 to 85 km, covering the period June 2002 to April 2012, has been created within the framework of the ESA funded MesosphEO project using the daytime Rayleigh scattering at limb observed by GOMOS.

Comparisons of the GOMOS temperature profiles with nighttime Rayleigh lidar temperature profiles measured at OHP show some differences which possess a vertical structure that may be partially explained by the contribution of the thermal diurnal tides. The GOMOS data set was used to build a temperature climatology as function of latitude and month. Subsequent comparison with the GOMOS external model has yielded an agreement which is better than 2 K in the upper stratosphere (below 48 km (1 hPa) where the model is driven by ECMWF), and between 5 to 10 K in the mesosphere (from 50 to 80 km where the model follows NRLMSISE-00 climatology). The evolution of the temperature at Equator shows the occurrence of temperature MILs with global longitudinal extension, descending in the period of approximately one month from 80 to 70 km. The equatorial climatology also shows a semi-annual temperature oscillation in the upper stratosphere, a stratopause altitude varying between 47 and 54 km, and an annual temperature oscillation in the mesosphere.

The technique outlined in this paper to derive temperature profiles from Rayleigh scattering at the limb can be applied to any other limb-scatter sounder observing in the spectral range 350-500 nm where the Rayleigh scattering is efficient and the absorption by ozone and other stratospheric

constituents are not overly important. The technique is also a good candidate for application to future missions involving small satellite constellations due to the simplicity of the principle.

**Glossary**

**AMSU**   Advanced Microwave Sounding Unit. 2

**ECMWF**   European Centre for Medium-Range Weather Forecasts. 1, 11, 12, 15

**ENVISAT**   ENVIronmental SATellite. 3, 4, 13

**ESA**   European Space Agency. 3, 15

**GNSS**   Global Navigation Satellite System. 2

**GOMOS**   Global Ozone Monitoring by Occultation of Stars. 1, 3–15

**GSWM**   Global Scale Wave Model. 9, 10

**MLS**   Microwave Limb Sounder on the Aura satellite. 3, 7

**NDACC**   Network for the Detection of Atmospheric Composition Changes. 3, 7

**OHP**   Observatoire de Haute Provence. 7–9, 15

**OSIRIS**   Optical Spectrograph and InfraRed Imager System. 3

**SABER**   Sounding of the Atmosphere using Broadband Emission Radiometry. 3, 7, 8

**SSU**   Stratospheric Sounder Unit. 2

**TIMED**   Thermosphere Ionosphere Mesosphere Energetics Dynamics. 3, 7, 8

**UARS**   Upper Atmosphere Research Satellite. 3, 7

**WINDII**   WIND Imaging Interferometer. 3

*Data Availability* GOMOS temperature profiles from Rayleigh scattering at limb are freely available at ESA MesosphEO Data product service: http://mesospheo.fmi.fi/data_service.html.

*Acknowledgements* This work was funded by European Space Agency (MesospEO project), Centre National d'Etudes Spatiales and CNRS/INSU.

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
