# Peer review of "A new MesosphEO dataset of temperature profiles from 35 to 85 km using Rayleigh scattering at limb from GOMOS/ENVISAT daytime observations"

_Atmospheric Measurement Techniques, 2018_

## Referee Comment (RC1) · Anonymous Referee #1 · 8 Oct 2018

The paper presents a new comprehensive Rayleigh stratosphere/mesosphere temperature profile dataset derived from the ESA ENVISAT GOMOS instrument. Paper is thus relevant and appropriate for publication in EGU's open access journal Atmospheric Measurement Techniques. Initial validation and first scientific results of a monthly climatology as compared to a corresponding climatology of ECWMF and NRLMSISE-00 is provided. A comparison with LIDAR measurements is also provided. Systematic differences are observed – though the differences are not unreasonable given the nature of the measurements and this nature of the MSIS climatology. Several hypotheses

regarding systematic differences (biases) with respect to the differences with climatology are given and explored – with the most likely being differences in local-time for LIDAR measurements, as well as Mie scattering by aerosols for the lower altitudes. The GOMOS Rayleigh scatter measurements, basically fixed in local-time, also exhibit so-called mesosphere inversion layers that have been observed by other research with earlier datasets. The data processing technique, which is based on techniques applied to earlier missions (e.g. Solar Mesosphere Explorer - SME) is noted and references are provided. With the information provided it should be possible for other scientists to reproduce the level 2-3 data products described from the GOMOS level 1 data products. Sufficient references to earlier research, and credit to earlier developments of the basic technique is given, with the specific details for GOMOS dataset processing provided.

The title and abstract provide a good description and overview of the paper. The abstract includes an overall summary of the initial scientific results some the secondarily derived month mean global climatology as well. The paper's presentation along with the equations, figures, and captions are straightforward and clear. Although an additional article supplement is not provided, a URL for a well-organized and easily accessible ESA project data-service to access and utilize the new dataset (along with other pertinent and corroborating ENVISAT datasets) is given.

With 309,000 temperature profiles from 35 to 85 km spanning about 10 years from 2002 to 2012 GOMOS Rayleigh scatter measurements will be a valuable resource to the middle-atmosphere research community. While a zonal monthly mean upper stratosphere and mesosphere, and related issues including tidal aliasing of local time limited observations is nothing new, the power of the dataset is in the ability to provide improved coverage in conjunction with other observations such as ground-based LIDARS, NASA TIMED SABER, NASA MLS, and other ESA ENVISAT instruments which measured temperatures in the same altitude region over approximately same time intervals. Such independent measurements improve statistics to better understand physical process in the region to better predict both day-to-day variability as well as seasonal and long-term behaviors of the region. With respect to the biases with respect to NRLMSISE-00 it will not surprising that the other recent ENVISAT and NASA SABER and MLS datasets show a similar result, indicating that NRLMSISE-00 probably needs to be updated.

The paper is basically acceptable for publication but as the authors may wish to add a few points of clarification for interested readers. First it might be worth mentioning in the paper that GOMOS had several other potentially independent means of making temperature profile measurements in the region as described in Bertaux et al., (2010), but that the Rayleigh scattering techniques seems the most reliable. Secondly, page 4 near line 30, the "et al., (2018a)" reference is missing. A comma is also missing on page 4 line 21.

---

## Referee Comment (RC2) · Anonymous Referee #2 · 9 Nov 2018

The study presents an interesting approach to retrieve temperature profiles in the middle atmosphere using the daytime Rayleigh scattering at limb from satellite observations (GOMOS/ENVISAT). Ten years of measurements have been used to obtain an impressive temperature dataset of more than 309000 profiles. As the authors pointed out, this dataset can be very useful for future climatology and atmospheric dynamics studies in the mesosphere.

A validation of these daytime temperature profiles has been done by means of a comparison with the nighttime Rayleigh lidar measurements. Some discrepancies were

found between both techniques and as the authors mentioned, they could be partially explained by the contribution of thermal diurnal tides. In this point I think it would have been interesting to compare also with other techniques (as for example microwave MLS measurements), in which the time difference between their measurements were lower than between lidar and GOMOS. It would have provided a better estimation of the accuracy of the GOMOS profiles. But I consider that it is something that can be addressed in future studies.

In addition, a temperature climatology has been built as function of latitude and month and compare with model. A more detailed analysis has been carried out for the temperature measurements at the equator, giving a good idea of the great potential of this database, which could lead to deeper analysis at a regional or global scale in future studies.

The paper is very interesting, well written and describe the potential of Rayleigh scattering at limb observations to gain temperature information in the middle atmosphere. I consider that this study is appropriated for Atmospheric Measurement Techniques and it should be ready for publication after correcting some typos and minor comments:

Minor comment:

- Pag 5, line 19: Indicate how many profiles are used for this statistics (validation using lidar observations).

Typos:

- page 4, line 2: Tukiainen et al -> add year of publication

- page 4, line 7: replace ".. is negligible" by ".. are negligible"

- page 4, line 26, . . .. noise) -> delete it

- page 4, line 29: in et al. (2018): the author is missing in the cite

- page 6, line 6: ". . .. for the 45°N latitude for August and middle panels)". Something

is wrong in this sentence.

---

## Author Comment (AC1) · 11 Dec 2018

Dear Reviewer #1

Thank you for your useful comments. Please find our answers below. Main comment The paper is basically acceptable for publication but as the authors may wish to add a few points of clarification for interested readers. First it might be worth mentioning in the paper that GOMOS had several other potentially independent means of making temperature profile measurements in the region as described in Bertaux et al., (2010),

but that the Rayleigh scattering techniques seems the most reliable. We agree with this comment. Bertaux et al. (2010) identified seven possible methods to determine temperature profiles from GOMOS data. Among them the two most promising are the vertical inversion of the Rayleigh scattering profile at limb presented in this study and the time delay between blue and red scintillations due to chromatic refraction, with an improved algorithm described in Sofieva et al. (2018). The two methods are complementary. The Rayleigh scattering method covers the altitude range 35-85 km during daytime and the chromatic refraction method covers the altitude range 15-32 km during nighttime. We added a paragraph in the revised version.

Minor comments Secondly, page 4 near line 30, the "et al., (2018a)" reference is missing. Corrected, the reference is Wing et al. (2018a). A comma is also missing on page 4 line 21. Corrected.

---

## Author Comment (AC2) · 11 Dec 2018

Dear Reviewer #2

Thank you for your useful comments. Please find our answers below.

**Main comment**

A validation of these daytime temperature profiles has been done by means of a comparison with the nighttime Rayleigh lidar measurements. Some discrepancies were found between both techniques and as the authors mentioned, they could be partially explained by the contribution of thermal diurnal tides. In this point I think it would have been interesting to compare also with other techniques (as for example microwave MLS measurements), in which the time difference between their measurements were lower than between lidar and GOMOS. It would have provided a better estimation of the accuracy of the GOMOS profiles. But I consider that it is something that can be addressed in future studies.

We agree that a comparison with other techniques observing the temperature at limb from space would be very useful. However this was beyond the scope of this study. The two most used space sensors for upper stratosphere – mesosphere temperature profiling are MLS-AURA and SABER-TIMED. These two sensors have been recently compared with the OHP Rayleigh lidars by Wing et al., (2018b) which showed systematic differences and suggested non-linear distortions in the satellite altitude retrievals. Despite the difference in local hour of measurements, GOMOS seems to be in better agreement with the OHP lidar at the stratopause region with less than 1 K bias, compared to nearly 4 K for SABER and greater than 8 K for MLS.

In order to better understand these differences, we plan to compare in a future work our new GOMOS temperature dataset with MLS and SABER. A comment is added on this point in the revised version.

**Minor comment**

- page 5, line 19: Indicate how many profiles are used for this statistics (validation using lidar observations).

554 collocated profiles have been compared. Added in the revised version.

**Typos:**

- page 4, line 2: Tukiainen et al -> add year of publication

Tukianen et al. (2011). Added.

- page 4, line 7: replace ".. is negligible" by ".. are negligible"

The sentence "... is negligible" seems OK for us.

- page 4, line  $26, \ldots$  noise) -> delete it

Sentence corrected.

- page 4, line 29: in et al. (2018): the author is missing in the cite

Wing et al. (2018a). Corrected.

- page 6, line 6: ". . .. for the  $45^{\circ}$ N latitude for August and middle panels)". Something is wrong in this sentence.

"and middle panels)" removed.